# First Molecular Detection of *Neospora caninum* in Feces of Grey Wolf (*Canis lupus*) and Golden Jackal (*Canis aureus*) Populations in Slovenia

**DOI:** 10.3390/ani13193089

**Published:** 2023-10-03

**Authors:** Petra Bandelj, Darja Kušar, Laura Šimenc, Urška Jamnikar-Ciglenečki, Gorazd Vengušt, Diana Žele Vengušt

**Affiliations:** 1Institute of Microbiology and Parasitology, Veterinary Faculty, University of Ljubljana, Gerbičeva ulica 60, 1115 Ljubljana, Slovenia; petra.bandelj@vf.uni-lj.si (P.B.); darja.kusar@vf.uni-lj.si (D.K.); laura.simenc@vf.uni-lj.si (L.Š.); 2Institute of Food Safety, Feed and Environment, Veterinary Faculty, University of Ljubljana, Gerbičeva ulica 60, 1115 Ljubljana, Slovenia; urska.jamnikar@vf.uni-lj.si; 3Institute of Pathology, Wild Animals, Fish and Bees, Veterinary Faculty, University of Ljubljana, Gerbičeva ulica 60, 1115 Ljubljana, Slovenia; gorazd.vengust@vf.uni-lj.si

**Keywords:** *Neospora caninum*, definitive hosts, grey wolf (*Canis lupus*), golden jackal (*Canis aureus*), real-time PCR (qPCR), digital PCR (dPCR)

## Abstract

**Simple Summary:**

*Neospora caninum* is an intracellular parasite that is the leading cause of reproductive failure in cattle worldwide and can also cause severe neuromuscular disease in dogs. The parasite circulates between herbivorous intermediate hosts (domestic and wild ruminants) and canine definitive hosts (e.g., dogs and wolves). The population of wild canids may play an important role in disease outbreaks in domestic ruminants, but it is poorly understood. Only definitive hosts shed the parasite in feces, thus samples from grey wolves and golden jackals were tested for the presence of *N. caninum* using a validated molecular method. The study confirmed a prevalence of 7.1% (3/42) in wolves and 2.6% (1/39) in golden jackals in Slovenia. This is the first molecular detection of the parasite in the population of grey wolves in Slovenia and the first detection in golden jackals. We suggest the golden jackal as a possible definitive host that may influence the spread of *N. caninum* in livestock.

**Abstract:**

*Neospora caninum* is an obligate intracellular parasite that causes reproductive disorders and major economic losses in cattle, and induces neuromuscular disorders in canids. Exogenous infections are becoming increasingly important due to disease outbreaks. The sylvatic life cycle of *N. caninum* interferes with the domestic dog-ruminant life cycle, but understanding of it is scarce. The population of wild canids may play an important role in parasite dispersion. Feces from 42 grey wolves (*Canis lupus*) and 39 golden jackals (*Canis aureus*) were analyzed for the *N. caninum* Nc5 gene using a novel real-time PCR (qPCR) with a detection limit of 5 targets/µL in clinical samples. Three wolves (3/42; 7.1%) and one golden jackal (1/39; 2.6%) tested positive, which is the first detection of *N. caninum* in the population of grey wolves in Slovenia and the first detection of *N. caninum* DNA in the feces of a golden jackal. In addition to the grey wolf, we propose the golden jackal as a potential definitive host with hypothetical epidemiological importance for the sylvatic-domestic life cycle of *N. caninum*, due to its proximity to human habitats and its rapid expansion throughout Europe.

## 1. Introduction

*Neospora caninum* is an obligate intracellular parasite from the Sarcocystidae family with domestic and wild canids as definitive hosts. Many warm-blooded animals, most commonly ruminants, are its intermediate hosts [1]. *N. caninum* was first observed in Norwegian dogs in 1984 [2] and described in 1988 as a new species distinct from *Toxoplasma gondii* or *Hammondia heydorni*, whose oocysts are morphologically identical [3]. In the domestic cycle, the parasite mainly circulates between domestic ruminants (cattle, small ruminants) and dogs, while in the sylvatic cycle it circulates between prey animals and mainly wolves. However, both cycles may be intertwined where livestock and wildlife share the same resources [4,5,6,7].

In canids, although rarely, *N. caninum* can cause neurological symptoms and death, especially in congenitally infected puppies, where it can form cysts in the central nervous system [8]. Infection occurs horizontally by ingestion of oocysts or vertically from the infected dam to the fetus. In cattle, the infection is of great clinical and economic importance, as *N. caninum* is considered as one of the most common etiological causes of failed pregnancies and stillbirths [9,10]. Birth of asymptomatically infected animals and congenital transmission play an important role in the persistence of the parasite within the infected herd [11,12]. A recent study reported transplacental transmission in 87.5% of *N. caninum*-infected free-ranging pregnant roe deer, wild boar and red fox [13]. Nevertheless, it remains questionable which species within the wildlife population play a significant role as definitive or intermediate hosts. Grey wolves (*Canis lupus*), coyotes (*Canis latrans*) from North America [5,14] and Australian wild dogs (*Canis lupus dingo*) [15,16] were confirmed as definitive hosts, and their prevalence ranges from 2.2% to 10% [6,14]. Red foxes (*Vulpes vulpes*) were suggested as possible definitive hosts [17], but research results are more consistent with their role as intermediate hosts [18,19,20]. Grey wolves in Slovenia are an indigenous species that was almost exterminated due to human persecution at the end of the 19th century [21]. Currently, the population of grey wolves is increasing and is part of the Dinaric Balkan population, which is known for its large size (about 5000–5500 individuals) and wide distribution [22]. Meanwhile, the greatest increase in population size and geographic distribution among carnivores in Europe has been observed in the golden jackal (*Canis aureus*) [23,24]. In Slovenia, golden jackals were first reported in the 1950s [24], with regular sightings since 2008 [25]. A broad home range, territorial mobility, a highly unselective diet and behavioral features predispose golden jackals to many infectious agents, including parasites. As noted with *Echinococcus multilocularis* [26], golden jackals may play a significant role in the dissemination of clinically important parasites, which warrants more extensive investigation. There is no report on the presence of *N. caninum* in their feces, although seropositive golden jackals were reported from Israel [27,28]. Sexual replication of *N. caninum* occurs in the gut of definitive hosts and oocysts are shed in their feces for a short period of time [29]. Thus, the use of a sensitive and specific real-time PCR (qPCR) assay to detect *N. caninum* is of paramount importance for epidemiological studies, especially in wildlife.

The aim of this study was to use a validated qPCR method calibrated with digital PCR (dPCR) to assess the prevalence of *N. caninum* in fecal samples from grey wolf and golden jackal populations in Slovenia, and to determine the role of the golden jackal in the life cycle of *N. caninum*.

## 2. Materials and Methods

### 2.1. Samples

Fecal samples from 42 grey wolves (*C. lupus*) and 39 golden jackals (*C. aureus*) were collected at the Institute of Pathology, Wild Animals, Fish and Bees (Veterinary faculty, University of Ljubljana, Slovenia). Animal carcasses of grey wolves and golden jackals were collected in 5 and 7, respectively, of 12 statistical regions in Slovenia (Figure 1 and Figure 2 in the Section 3) as part of the regular annual hunting bag or from road kills throughout the Slovenian territory from 2013 to 2020. All samples were stored at −80 °C for at least one month prior to processing and DNA extraction to avoid human exposure to viable parasite eggs of zoonotic importance. All samples were collected postmortem, thus ethics committee/welfare authority approval was not required.

### 2.2. Methods

#### 2.2.1. Design of *N. caninum* qPCR Assay

For qPCR, we selected the repetitive Nc5 gene for *N. caninum* [30]. It is highly specific and is one of the two most common markers used for routine PCR-based *N. caninum* detection [31]. Since we were aiming for a calibrated qPCR for absolute quantification and also confirmation of amplicon identity after qPCR by Sanger sequencing, we constructed a novel TaqMan assay targeting a similar but longer Nc5 region than previously reported [32]. The primers and probe were designed using the Primer Express v3.0.1 software (Applied Biosystems by Thermo Fisher Scientific, Waltham, MA, USA). Nc5 gene sequence alignment was performed using the Nc5 hybridization probe region from strain NC-1 (GenBank accession number X84238.1) as a template [33]. Primer specificity was tested with the NCBI Primer BLAST tool (http://www.ncbi.nlm.nih.gov/tools/primer-blast/, accessed on 11 July 2023) using the standard nr nucleotide database. The *N. caninum* specific TaqMan assay (Table 1) generated a 102 bp amplicon.

#### 2.2.2. Validation and Calibration of qPCR

The qPCR reaction mix contained 300 nM of both primers and 100 nM of the probe (Table 1). When implementing the qPCR protocol, the optimal total volume of DNA/PCR mix was set at 2/25 µL. During validation, the Maxima Probe qPCR Master Mix (2×) with separately added 1:10 ROX solution (Thermo Fisher Scientific, USA) showed better performance and no inhibition compared to the FastStart Universal Probe Master (Rox) (Roche, Basel, Switzerland). qPCR was performed on the QuantStudio 5 Real-Time PCR System (Applied Biosystems by Thermo Fisher Scientific, USA) with thermal cycling conditions consisting of a preheating step at 50 °C for 2 min, followed by 95 °C for 10 min and 45 cycles of denaturation at 95 °C for 15 s, with annealing and extension at 60 °C for 1 min. After validation, the threshold line was set at 0.1 for all samples. Positive and negative (PCR-grade water) controls were included in each qPCR run. Results were expressed in quantification cycle (Cq) values.

Positive control for *N. caninum* was provided by the Portugal National Institute (Instituto Nacional de Investigação Agrária e Veterinária, I.P.). To detect and quantify the *N. caninum* Nc5 genomic target, linear regression of a standard curve was performed to validate qPCR. The positive control was diluted in DNA extracted from *N. caninum*-negative animal feces in a 5-fold series to mimic the *N. caninum*-positive feces with decreasing amounts of *N. caninum*; this was used to prepare the qPCR dilution series to account for the impact of PCR inhibitors that may be present in the animal feces. Each dilution was qPCR tested in three technical replicates. For the calculation of a linear regression equation, only data within the linear dynamic range (LDR) were considered. This is defined as dilutions with the coefficient of variation (CV) not exceeding 33%, since CV markedly increases below the limit of quantification (LOQ) [34], which indicated the lower limit of LDR. The limit of detection (LOD) was determined accordingly, being 5- to 10-times lower than LOQ in complex samples [35]. For determination of the Cq cut-off value, the last standard dilution where both positive and negative results were observed was considered; here, the highest Cq value was rounded up to the next half value and increased by 0.5 [36]. The qPCR amplification efficiency was calculated according to the equation E = 10^−1/slope^ − 1 [34].

For calibration of qPCR with dPCR, the same TaqMan assay and the prepared DNA dilution series were employed as for qPCR. The reaction mix contained 7.5 µL of 2× QuantStudio 3D Digital PCR Master Mix v2 (Applied Biosystems by Thermo Fisher Scientific, USA), 300 nM of both primers and 100 nM of the probe, 3 µL of DNA, and PCR-grade water to a final volume of 15 µL. The prepared dPCR reactions were loaded onto QuantStudio 3D Digital PCR 20K Chips v2 (consisting of 20,000 reaction wells per chip) and amplification was performed on a ProFlex 2× Flat PCR System (Applied Biosystems by Thermo Fisher Scientific, USA) according to the manufacturer’s instructions; the same number of cycles and annealing temperature was used as for qPCR. Chip imaging was performed with the QuantStudio 3D Digital PCR Instrument and results were analyzed using the QuantStudio 3D AnalysisSuite v3.1.6 Cloud Software. Results of dPCR quantification were expressed in copies of Nc5 per µL of the extracted DNA and were used for calibration of qPCR; by analyzing the same samples (DNA dilution series) by dPCR and qPCR, Cq values from qPCR validation could be translated into absolute quantification values and validation parameters of qPCR determined.

#### 2.2.3. DNA Extraction from Animal Feces

DNA extraction from fecal samples was performed using the iHelix kit (Institute of Metagenomic and Microbial Technologies, Slovenia; https://www.ihelix.eu/, accessed on 11 July 2023) according to the manufacturer’s instructions and as previously described [37]. Prior to extraction, each fecal sample was diluted (approx. 0.5 g of fecal sample with 1 mL of sterile saline solution) in the provided 2-mL screw cap tubes and vortexed to obtain a homogenized solution. The extraction protocol included bead-beating (45 s at 6400 rpm) three times using a tissue homogenizer (MagNA Lyser Instrument; Roche, Switzerland), combined with enzymatic and heat-induced lysis between mechanical shearing. DNA was eluted to a final volume of 100 µL and stored at −20 °C till qPCR analysis. DNA from all samples was diluted (1:10) and retested to rule out possible inhibition.

## 3. Results

### 3.1. Validation and Calibration of qPCR

qPCR for the detection of *N. caninum* Nc5 genomic target in animal fecal samples showed good performance with a LOQ of 8 targets per µL of the extracted DNA and a LOD of 5 targets per µL, reaching the reported theoretical limit of three target copies per PCR [38]. The equation for the linear regression curve was y = −3.0978x + 37.313 with a regression coefficient R^2^ of 0.9984. The efficiency of qPCR was 110.3%. The Cq cut-off value was set at 41.0.

### 3.2. N. caninum in Fecal Samples from Grey Wolfs and Golden Jackals

Of 42 grey wolf and 39 golden jackal fecal samples, three grey wolves (3/42; 7.1%) and one golden jackal (1/39; 2.6%) tested positive for *N. caninum* DNA by the newly developed and validated qPCR. The Cq values of *N. caninum*-positive animals were 38.1, 38.2 and 39.3 for grey wolves (one female, two males; 1, 5 and 8 years old) and 40.1 for the golden jackal (female, juvenile less than 1 year old). Positive results were confirmed with Sanger sequencing; all sequences were most closely related to the sequence *N. caninum* Liverpool complete genome, chromosome II (GenBank accession number FR823382.1).

Two of 25 (8.0%) samples from grey wolves collected in the Slovenian region R3 and one of six (16.7%) from region R6 were positive for *N. caninum* (Figure 1). One of 9 (11.1%) samples from golden jackals in region R7 was also positive (Figure 2).

**Figure 1 animals-13-03089-f001:**
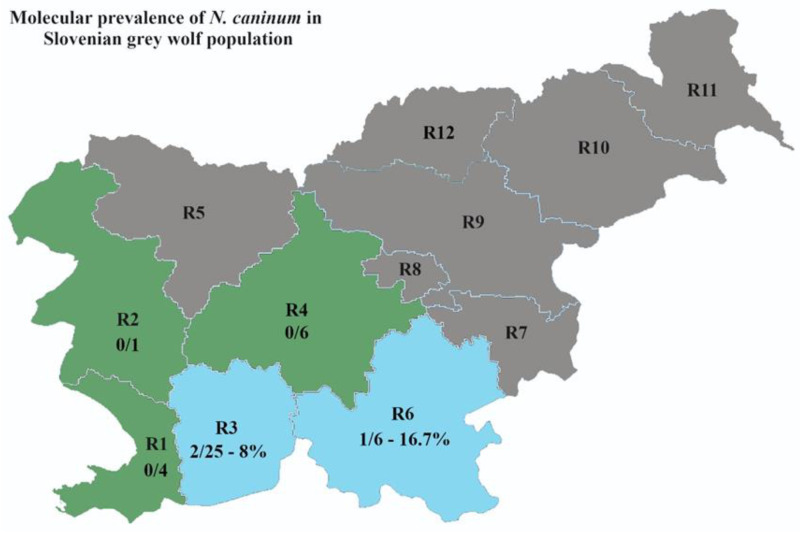
Molecular prevalence of *Neospora caninum* in the population of grey wolf (*Canis lupus*) in Slovenian regions. Regions (R): R1, obalno–kraška; R2, goriška; R3, primorsko–notranjska; R4, osrednjeslovenska; R5, gorenjska; R6, jugovzhodna Slovenija; R7, posavska; R8, zasavska; R9, savinjska; R10, podravska; R11, pomurska; R12, koroška. *N. caninum* sampling regions: positive in blue, negative in green, without samples in grey.

**Figure 2 animals-13-03089-f002:**
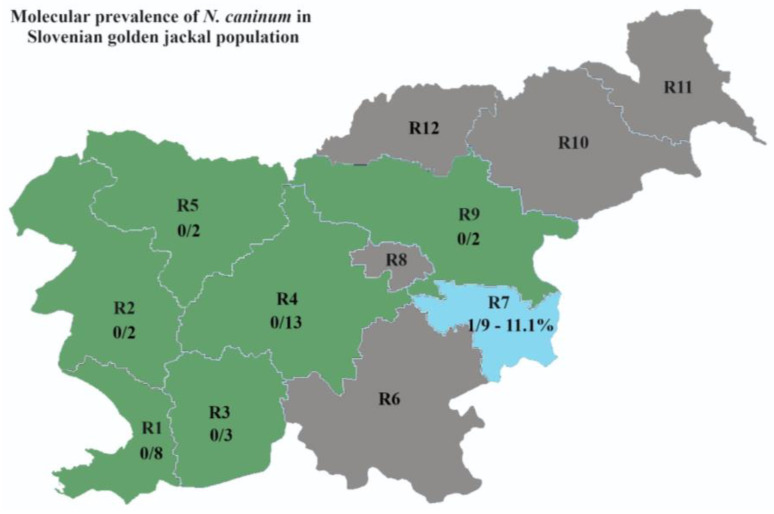
Molecular prevalence of *Neospora caninum* in the population of golden jackal (*Canis aureus*) in Slovenian regions. For region designation and color legend, see Figure 1.

## 4. Discussion

*N. caninum* is a major cause of economic losses in the cattle industry due to reproductive failure in dams [2,9,10]. The parasite can transmit vertically, which plays an important role in its persistence within the herd [13,39]. However, exogenous exposure to oocysts is also important, especially in disease outbreaks [40,41]. The sylvatic life cycle of *N. caninum* occurs in parallel with the domestic dog-ruminant life cycle, but understanding of it is scarce [1]. In this study, we determined the molecular prevalence of *N. caninum* in the Slovenian grey wolf population and for the first time detected *N. caninum* DNA in the feces of golden jackals.

The confirmed definitive hosts of *N. caninum* in the sylvatic cycle are grey wolves [6], coyotes [14] and Australian wild dogs [15,16]. The reported seroprevalence in grey wolves ranged from 3.2% to 39.0% [42], while a study from the United States found a molecular prevalence of 4.1% in wolf feces [6]. The prevalence observed in our study was similar, with 7.1% *N. caninum* positive grey wolves. In neighboring countries, there are no molecular data available for the presence of *N. caninum* in grey wolves. A recent study from Italy reported the presence of *Neospora*/*Hammondia* spp. oocysts in only one fecal sample (1.3%) [43], while in Croatia their presence was confirmed for 2.6% fecal samples using non-molecular coprological methods [44]. Both studies could benefit from the use of molecular methods to determine the prevalence of *N. caninum* in their samples. Recent data indicate that the Slovenian grey wolf population has increased from 34–42 in 2010/2011 to 120 (106–147) individuals in 2020/2021 and an expansion of its territory has been observed, which is to be expected as the grey wolf in Slovenia is part of a large and viable Dinaric Balkan population. [22,45,46]. As expected, all *N. caninum*-positive grey wolves in our study originated from the Dinaric karst region or its vicinity (regions R3 and R6 in this study, Figure 1), where the highest grey wolf population densities in Slovenia have been recorded [45]. The occurrence of wolf-dog hybrids is also of concern [45], as the animals may be less cautious and approach human habitats, favoring parasite dispersion. In addition, cases of predators attacking small ruminants and even cattle are increasing [45]. Although wolves rarely prey on adult cattle [45], they could still defecate on grasslands and pastures where grass or eventually hay would be eaten by ruminants. Even a low prevalence of *N. caninum* must be considered, since *N. caninum* oocysts are extremely resistant in the environment and a small number of oocysts have been shown to be sufficient to infect an intermediate host [41,42]. Globally, the pooled molecular prevalence of *N. caninum*-aborted fetuses was reported to be 15% in small domestic ruminants [12] and 43% in cattle [11]. In Italy, Zanet and colleagues reported a prevalence of *N. caninum*-positive fetuses of 31% in roe deer and 25% in wild boars [13]. Studies in wildlife have often focused on intermediate hosts and their role in the life cycle of *N. caninum* [13,47]. However, there is still no clear evidence of the impact of *N. caninum* infection on the health of individual animals or wildlife populations [47].

On the other hand, the rapidly growing population of the golden jackal in Slovenia and in Europe should be taken into consideration when investigating their role in the life cycle of *N. caninum* [24,48]. Due to the expansion and the nature of their behavior, they come close to the rural habitat of humans and often prey on small domestic animals in addition to rodents [49]. The seroprevalence of *N. caninum* in golden jackals has been reported to range from 1.7% to 3.2% in Israel [27,28], suggesting that golden jackals are exposed to the parasite. Because only seroprevalence data are available, no assumptions could be made on the role of golden jackals in the life cycle of the parasite. In this study, we report for the first time the presence of *N. caninum* DNA in the feces of a golden jackal. With a prevalence of 2.6%, it is in close concordance with the Israeli study, although a comparison should not be made due to differences in methodology [14]. However, a report from a Canadian study showed agreement with the 10–18% seroprevalence in coyotes from the USA studies and a 10% molecular detection of oocysts in the feces of Canadian coyotes [14,42]. Both species of wild carnivores included in our study showed a low prevalence, which may suggest that neither play an important role in the epidemiology of *N. caninum*. However, based on the recent Italian study and considering the increase in presence and density observed for both species in Slovenia over the last decade [23,24,45,48], their role may still be evolving [13]. What should also be considered is that canids presumably spread *N. caninum* only from day 5 and up to day 30 post infection [42]. Thus, it is challenging to find positive wild animals that actively shed *N. caninum* oocysts in their feces. For this reason, we used a very sensitive and specific molecular tool to optimize and possibly expand the three-week shedding window of *N. caninum*.

The obtained LOD value of the constructed qPCR assay (5 Nc5 copies per µL of extracted DNA) indicates that the assay can detect down to 500 Nc5 targets per 0.5 g of animal feces. However, this does not correlate directly to the number of *N. caninum*, as the Nc5-type sequences have a high redundancy within the genome of the parasite [50]. The good performance of the constructed and validated qPCR enabled us to identify positive animals even at the extreme end of the reliable detection limit of the method. On the other hand, the amount of *N. caninum* DNA in positive samples was not sufficient for sequencing larger DNA segments for strain typing or attempting to determine the viability of the oocysts. Counting the *N. caninum* oocysts would have enhanced the scientific value of the study, but it was not possible due to the lack of sample material and storage at −80 °C to prevent zoonotic infections with *Echinococcus* sp. Due to these shortcomings, we can only speculate that golden jackals may be definitive hosts of *N. caninum*. However, since golden jackals are considered functional “cleaners” in human-dominated landscapes by consuming discarded animal waste [51], they could acquire *N. caninum* from discarded aborted fetuses. This, in turn, would mean that the potential role of golden jackals as definitive hosts, with their rapid population increase and expansion throughout Europe, could increase the risk of infection for livestock animals raised at the interface with free-ranging wildlife. Previous findings from North America confirmed horizontal transmission of *N. caninum* infection between wild and domestic animals, where naturally infected deer transmitted *N. caninum* to domestic dogs after dogs were fed infected deer tissue and subsequently shed oocysts [52]. In this study, we report the confirmed presence of *N. caninum* in two common species of wild canids. To date, there are no data on the occurrence of this parasite in domestic dogs or livestock in Slovenia. However, it is a common hunting practice in Slovenia that wild ruminants, e.g., roe deer and red deer, are field dressed and the offal is left exposed not only to wild canids but also to domestic dogs (e.g., sheepdogs and hunting dogs) in the wild, which poses a potential risk of the transmission of *N. caninum* to livestock. This should encourage further studies focused on defining the relevant players in the *N. caninum* sylvatic-domestic life cycle.

## 5. Conclusions

In conclusion, *N. caninum* is present in the feces of the Slovenian population of grey wolves and golden jackals. This is the first report on molecular detection of *N. caninum* DNA in the feces of a golden jackal, supporting its role as a possible definitive host of the parasite. The newly described qPCR is specific and sensitive enough to be used in epidemiological studies for *N. caninum*. Further studies are needed to determine the exact role of the definitive and intermediate hosts in the sylvatic-domestic life cycle of *N. caninum*.

## Figures and Tables

**Table 1 animals-13-03089-t001:** *Neospora caninum* TaqMan assay used in this study.

Primer/Probe	Oligonucleotide Sequences (5′–3′)
NC forward primer	GGAGGACATCGCTCACTGAC
NC reverse primer	GCTCCACCAACAATGCTTCG
NC probe	[FAM]–AGGCACGCTGAACACCGTATGTC–[TAMRA]

## Data Availability

The data presented in this study are available upon request from the corresponding author.

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
