# Peer review of "First Molecular Detection of Neospora caninum in Feces of Grey Wolf (Canis lupus) and Golden Jackal (Canis aureus) Populations in Slovenia"

_animals, 2023, doi:10.3390/ani13193089_

Round 1
Reviewer 1 Report
Dear Authors,
I read with interest the manuscript, as the epidemiology of Neosporosis in the wild environment is still partially unknown and deserve some more investigations.
I have one major concern about the method, and I am asking if it would be possible to complete the study with a microscopic examination (and consequent oocysts count) on the stored stool samples, at least the positive ones, to:
- confirm at the microscope the presence of intact (presumably viable at the moment of sampling) Neospora oocysts and try to quantify the parasitic load;
- evaluate the presence of some other Apicomplexa parasites that could possibly interfere with the diagnosis, affecting the specificity of the molecular method. Analytical specificity tests are not described in the qPCR validation, even if the specificity is guaranteed, as the positive results were confirmed with Sanger sequencing.
Apart from lack of specificity (I do not think this is the problem), the use of a very sensitive method could lead to false positive results, due to a possible passive contamination of carnivores’ feces caused by ingestion of environmental oocysts and their passive transition through the carnivores’ gut. The molecular method, although very sensitive, do not allow a quantification of the parasitic charge and could detect small genome amounts of non-viable parasites.
I read that the stool samples were stored at -80°C, so some oocysts will probably be broken, but the chance to highlight some whole Neospora oocysts and to count them (with a Mc Master chamber) could increase the scientific value of the study results.
Moreover, the calculated confidence intervals seem to be too strict: please specify the CI (95%?).
Some minor comments:
Line 20 and through all the manuscript: please correct the repeated typing error “7.1 %” to 7.1% (no spacing between numbers and “%”).
Line 22: I suggest changing “We suggest the golden jackal as a possible definitive host that may greatly influence the spread of N. caninum in livestock” to “We suggest the golden jackal as a possible definitive host that may influence the spread of N. caninum in livestock” (more prudent)
Line 34: “we propose the golden jackal as a potential definitive host with high epidemiological importance” I would change to a lighter statement, such as “we propose the golden jackal as a potential definitive host with a hypothetic epidemiological importance” (more prudent)
Line 62: is the cited prevalence referred to the shedding status? Or to the status of definitive host? Please specify
Line 174: The Cq cut-off value was set at 41.0 based on which data? Do the Authors know to how many oocysts/g do this value correspond?
Line 224-226: some sentences are repeated, maybe due to the revision (In addition, cases of predators attacking small ruminants and even cattle are increasing [46]. Although wolves rarely prey on adult cattle, because they generally prey on smaller animals, such as deer or sheep [48])
Line 229: I do not think appropriate the definition “resilient” for oocysts. I would think better “resistant”.
Line 261-262: as declared by the Authors, the high redundancy of Nc5-type sequences does not allow a real quantification and seems to be inconsistent with the goal declared at line 100 (absolute quantification). Maybe I did not well understand the method description.
Reviewer 2 Report
The manuscript submitted by Bandelj et al. presents some really interesting results on the presence of Neospora caninum in grey wolf and golden jackal feces. In addition, they describe a new method for detecting the parasite using qPCR.
The writing of the manuscript is very good, both the introduction and the discussion contain appropriate references to the topic, the methodology describes all the analyses step by step very well and the results are presented in a very clear and easy-to-read way.
I would have liked the presentation to be a little more careful (parts of the text with track changes and paragraphs without merge), but I have not detected any major errors.
I only have a couple of suggestions for the authors, which is really only one. In the collection of samples, Slovenia was divided into 12 statistical regions. What criteria were used to divide these regions? Was it based on the ecosystem, the climate, or geography? I think it would be interesting to indicate this, and even see if the regions with positive individuals are positive due to bioclimatic differences that could affect the transmission of the parasite. Additionally, Figures 1 and 2 showing the arrangement of these regions are in the results. Perhaps it would be interesting to move them to this section, or simply add a map without the results, just with the division of regions.
Author Response
Dear Reviewer,
Please see our response attached.

Reviewer 3 Report
All comments and suggestions (mostly linguistic ones) have been posted on the attached file. In addition, for the benefit of readers, I think that Authors should try to improve the appeal of the Discussion. In particular, I invite them to recall knowledge and hypotheses on the Neospora sylvatic cycle , and discuss the infection risk for livestock raised at the interface with free-ranging wildlife

A moderate revision of the English is recommended
Author Response
Dear reviewer,
Please see our response attached.

Round 2
Reviewer 1 Report
All comments received adequate response. I would only reccommend, if the Authors do agree, to add in the discussion that "finding/counting the N. caninum oocysts would have enhanced the scientific value of the article. However, due to the lack of sample material and storage at -80°C to prevent zoonotic infections with Echinococcus sp., it has been not possible to perform these methods."
Author Response
Changes are highlighted in the document.